# Oxidation Behavior of FeNiCoCrMo_0.5_Al_1.3_ High-Entropy Alloy Powder

**DOI:** 10.3390/ma17020531

**Published:** 2024-01-22

**Authors:** Anton Semikolenov, Mikhail Goshkoderya, Tigran Uglunts, Tatyana Larionova, Oleg Tolochko

**Affiliations:** 1Institute of Machinery, Materials and Transport, Peter the Great St. Petersburg Polytechnic University, St. Petersburg 195251, Russia; semikolenov.a@edu.spbstu.ru (A.S.); uglunts.t@edu.spbstu.ru (T.U.);; 2Institute of Laser and Welding Technologies, State Marine Technical University, St. Petersburg 190121, Russia; 3The Federal State Unitary Enterprise “Central Research Institute of Structural Materials “Prometey”, Named by I.V. Gorynin of National Research Center “Kurchatov Institute”, St. Petersburg 191015, Russia; gosmike@yandex.ru

**Keywords:** high-entropy alloy, gas atomization, high-velocity oxygen fuel, oxidation, microstructure

## Abstract

One of the most promising applications of FeNiCoCrMoAl-based high-entropy alloy is the fabrication of protective coatings. In this work, gas-atomized powder of FeNiCoCrMo_0.5_Al_1.3_ composition was deposited via high-velocity oxygen fuel spraying. It was shown that in-flight oxidation of the powder influences the coating’s phase composition and properties. Powder oxidation and phase transformations were studied under HVOF deposition, and during continuous heating and prolonged isothermal annealing at 800 °C. Optical and scanning electron microscopy observation, energy dispersive X-ray analysis, X-ray diffraction analysis, thermogravimetric analysis, differential thermal analysis, and microhardness tests were used for study. In a gas-atomized state, the powder consisted of BCC supersaturated solid solution. The high rate of heating and cooling and high oxygen concentration during spraying led to oxidation development prior to decomposition of the supersaturated solid solution. Depleted Al layers of BCC transferred to the FCC phase. An increase in the spraying distance resulted in an increase in α-Al_2_O_3_ content; however, higher oxide content does not result in a higher microhardness. In contrast, under annealing, the supersaturated BCC solid solution decomposition occurs earlier than pronounced oxidation, which leads to considerable strengthening to 910 HV.

## 1. Introduction

High-entropy alloys (HEAs) have been already the focus of attention and study for almost two decades [1,2,3,4,5,6,7,8,9]. During this period of study, the original concept of HEA as a one-phase multicomponent solid solution stabilized by high entropy has been expanded and in a great extent shifted to numerous multiphase multiprincipal alloys, which generally possess superior strength. HEA based on 3d transition metals Fe, Ni, Co, and Cr still are the most studied HEAs and nowadays the most suitable for industrial application [10,11,12]. When Al is added in a content exceeding about 15%, the body centered cubic (BCC) crystal structure becomes predominant [9,10,11,12]. BCC HEA possess high hardness, thermal stability, excellent wear, and corrosion resistance [9,10,11,12]. For example, BCC AlCoCrFeNiMo_0.5_ alloys display a higher hot hardness level than those of Ni-based superalloys up to 1000 °C and have better softening resistance [9]. Thus, Al_x_CoCrFeNiMo_0.5_-based alloys have a great potential in applications at elevated temperatures and can be substitutes for Ni-based superalloys. 

One of the most promising applications of AlCoCrFeNi HEA is the fabrication of protective coatings [13,14]. HEA-based coatings, deposited on medium-carbon steel substrates, can considerably enhance the material performances, extend service life, and reduce product costs [13,14,15]. Thick coatings with low porosity and a high level of adhesion can be produced via the high velocity fuel spraying (HVOF) technique [13,15]. HVOF is a combustion spray process in which a hydrocarbon fuel (kerosene, methane, propane, etc.) is ignited, mixing with oxygen to accelerate feedstock powder towards the substrate [13,15]. Depending on the particle’s size and location in the high-temperature flow, the particles undergo various transformations, such as melting, softening, spheroidization, precipitation, and surface or total oxidation, similar to those observed with the atmospheric plasma spraying (APS) method [14]. Lower particle temperatures than those achieved in plasma spraying and higher particle speeds result in lower in-flight particle oxidation and better coating densities. 

As reported in [16,17,18,19], HVOF as-sprayed AlTiCrFeCoNi coatings possessed a dense microstructure and a high microhardness over 750 HV with a good fracture toughness. The heat-treated NiCoFeCrSiAlTi coating exhibits a hardness even higher than 800 HV, good wear resistance, and oxidation resistance at 1100 °C, similar to that of MCrAlY coating [18]. The cavitation erosion resistance of the AlCoCrFeNi coating is about 3.5 times that of 06Cr13Ni5Mo steel in the same environment [20]. The high wear resistance of AlCoCrFeNi-based HEA at elevated temperatures is associated with oxidation behavior. The formation of a compact oxide layer on the AlCoCrFeNi and AlTiCrFeCoNi HVOF coatings accounts for enhanced wear resistance at temperatures ≥ 800 °C [17] and a lowered coefficient of friction due to its function as a solid lubricant [19]. In [21], it was found that additions of Nd and Mo to a AlCrFeCoNi base significantly reduced the wear depth at increasing temperatures due to improved oxide layer adhesion. Thus, AlCoCrFeNiMo coatings deposited using HVOF are suitable for high-temperature application. 

A unique feature of thermal spray processes is the extremely high heating and cooling rates (>10^6^ K/s) [13,14,15]. Such rapid cooling rates result in a fine-grain and non-equilibrium structure due to the prevention of elemental segregation [13,14,15]. The feedstock powders most commonly used for thermal spraying are highly nonequilibrium structures produced by mechanical alloying or gas atomization that undergo phase changes when heated [22,23,24]. Therefore, as-sprayed HEA coatings may possess different phase constitution compared with their as-cast counterparts [24] as well as with the feedstock powder [22,23]. For example, both as-cast AlSiTiCrFeCoNiMo_0.5_ and AlSiTiCrFeNiMo_0.5_ are dual-phase alloys (BCC + FCC); however, if fabricated via thermal spray, they have only the supersaturated BCC solid solution [25]. Cold-sprayed AlFeNiCoCr alloy retains the same BCC structure as was observed in the powder [26]. In contrast, in the plasma-sprayed coating of the same composition, the FCC phase is observed as the major one [27]. Despite high heating and cooling rates, which prevent the elements’ diffusion development, factors such as high temperature, high oxygen concentration, and high specific surface of the powder inevitably lead to in-flight oxidation. The existence of oxides in the flame-sprayed HEA coatings due to in-flight oxidation improves the hardness and wear of the coatings compared to those of cold-sprayed ones [28]. A comparative study on FeCoCrNiMo_0.2_ HEA coatings produced using HVOF and APS shows that a higher fraction of spinel oxides in the APS HEA coating considerably reduced their wear rates compared to that in the HVOF-deposited coating [29]. This was mainly attributed to the self-lubricated effect caused by the oxide.

Thus, in-flight oxidation of the AlCoCrFeNi particle during the spraying process can be beneficial for the tribological performance of the HEA coatings. Due to high specific surface area, the powder can oxidize differently from the cast bulk alloy. Moreover, the formation of multiple oxides—alumina, chromia, and spinel—changes the residual particle’s core composition and can lead to phase destabilization. Therefore, understanding the regularity of the oxidation behavior and phase transformation of the high-entropy powder at different conditions should be necessary for the production of a coating with high and determined property levels.

In this work, the oxidation behavior and phase transformations of FeNiCoCrMo_0.5_Al_1.3_ high-entropy alloy powder, produced via gas atomization, were studied. Powder oxidation and phase transformations were studied under HVOF deposition, during continuous heating, and under prolonged isothermal annealing at 800 °C. FeNiCoCrMo_0.5_Al_1.3_ HVOF coatings were produced using various distances from the spray nozzle to the substrate to produce coatings with different oxide contents, and the coatings’ properties were investigated.

## 2. Materials and Methods

FeNiCoCrMo_0.5_Al_1.3_ alloy powder was produced via gas atomization; the parameters of the process are described elsewhere [24]. The chemical composition of the powder is presented in Table 1.

The coatings were deposited using high-velocity oxygen fuel (HVOF) using Hipojet 2700M, MP-2100, PF-3350 (Metal Spray Coating Corp., Hollywood, CA, USA). The spraying parameters are presented in Table 2. Structural carbon steel (grade of 09G2S) substrates of 25 × 90 mm size with a thickness of 8 mm were used. The surface of the substrates was previously subjected to sandblasting with electrocorundum with a particle size of 100 μm.

The phase composition of the coatings was examined with the D8 Advance diffractometer (Bruker, Billerica, MA, USA) using Cu Kα radiation. The solid solution lattice parameter was determined on the basis of XRD patterns, which had been collected in the 2Θ range from 20° to 140° with a speed of 2°/min. 

Optical microscopy was performed using Meiji IM7400 (Meiji Techno, Saitama, Japan). Scanning electron microscopy (SEM) was performed using a MIRA 3 (TESCAN, Brno, Czech Republic) microscope with the AztecLive.Advanced.Ultim.Max.65 energy-dispersive X-ray spectroscopy (EDX) detector (Oxford instruments, Abingdon, UK). In order to determine the volume fraction of the pores and oxides, the images of the microstructures were analyzed via the software THIXOMET Pro (Thixomet, Saint-Petersburg, Russia). Vickers microhardness was tested using a FM-310 (FUTURE-TECH Corp., Kanagawa, Japan) under a load of 50 g and a dwelling time of 10 s, and at least 7 measurements per point were made. 

DTA and TGA curves of the powder were obtained after heating at 20°/min for up to 900 °C in air using a Simultaneous Thermal Analyzer STA 449 F (Netzsch, Hanau, Germany). X-ray diffraction (XRD) of the samples under heating was carried out using the multifunctional X-ray diffractometer Rigaku Ultima IV (Rigaku Corporation, Tokyo, Japan) in a high-temperature chamber using Cu Kα-radiation. 

## 3. Results and Discussion

### 3.1. Powder Feedstock Characterization

The characterization of the feedstock powder is shown in Figure 1. The powder particles have a spherical shape (Figure 1a) with a maximum particle diameter distribution of about 20–40 μm (Inset of Figure 1a). In spite of the distinguished contrast in back scattering electron mode SEM image indicating element segregation (Figure 1b), there was only one phase—BCC solid solution—detected using XRD (inset of Figure 1b). A small peak at 30° indicates some BCC ordering, predominantly AlNi B2 [10,24].

### 3.2. Oxidation and Phase Transformations in the Powder during HVOF Spraying

The microstructures of the coatings manufactured using a different distance from the HVOF nozzle to the substrate are shown in Figure 2.

The coatings have a layered microstructure typical for coating produced using HVOF. In general, the large particle’s bodies have a flattened shape as a result of strike to the substrate in a molten or softened state [13]. The thin-layered structure observed between the large particles consists of two phases, which are different in element composition (the dark one is enriched with Al and the light one is depleted in it). 

Phase compositions of the powder and the coatings were analyzed with XRD, the results are shown in Figure 3a. Unlike the as-atomized powder, the phase composition of the coatings includes, in addition to BCC solid solution, FCC phase and alumina. A weak B2 peak can be noticed. The elemental compositions of the observed phases determined with EDX are presented in Table 3. Figure 3c shows the schematic microstructure of the coating.

Figure 4 shows the difference in the properties based on the spraying distance. As the spraying distance increases, the average thickness of the coatings slightly decreases due to a stronger scattering of the particles on the longer path. An increase in the spraying distance results in an increase in coating porosity. Porosity depends on the temperature and kinetic energy of the particles [30]. Particles sprayed from a longer distance have a lower kinetic energy when they strike the substrate, which results in increased porosity. High temperatures and exposure to oxygen lead to the in-flight oxidation of particles; the smaller particles undergo more severe oxidation. With an increase in the spraying distance, the particles are exposed to oxygen for a longer time, which leads to an increased content of oxides (Figure 4d). As the particles oxidize with the formation of alumina, BCC solid solution loses aluminum, which is reflected in a decrease in its lattice parameter (Figure 3b). It is known [10] that when Al concentration in FeCoNiCrAlMo_0.5_ is lower than 10%, FCC solid solution becomes preferable to BCC. So, in the areas close to the surface, the Al-depleted BCC becomes unstable and transforms to FCC.

As seen from Figure 4b, the higher oxide content in the coating deposited from a higher distance does not result in a higher microhardness. On the contrary, with an increase in the spaying distance from 150 mm to 250 mm, the microhardness decreases from 750 HV to 550 HV. Taking into account earlier observations, a noticeable decrease in the microhardness with an increase in the spraying distance can be explained by the increased porosity, the increase in the amount of the soft FCC phase, and the softening of the BCC solid solution due to Al outflow [10].

Thus, particle oxidation during high-temperature spraying in an oxygen-containing atmosphere considerably influences the coating properties. 

### 3.3. Oxidation and Phase Transformations in the Powder during Continuous Heating

Furthermore, the oxidation behavior and phase composition of FeNiCoCrMo_0.5_Al_1.3_ were studied during continuous heating in an air atmosphere. Figure 5 shows the DTA curve and rate of mass change of the powder during heating in air from room temperature to 1400 °C. For convenience, the TGA result is presented as a derivative (dm/dT). Therefore, three temperature intervals of transformations can be distinguished: The first one is up to 400 °C, where no significant effect is observed. The second one is 400–800 °C, where DTA curve deviation from the horizontal is not accompanied by mass change, so it is caused only by supersaturated solution decomposition. The third one is over 800 °C, where the DTA curve resembles the TGA derivative one, having the same mode and extremum positions, which can be attributed to complex oxidation processes.

In situ XRD patterns of the powders heated to different temperatures are shown in Figure 6. Up to 950 °C, only the decomposition of the BCC solid solution with sigma formation occurs, and the B2 peak becomes more pronounced. Sigma reflexes start to appear at 650 °C and intensify as the temperature rises to 950 °C. Reflexes of α-Al_2_O_3_ become more noticeable at 1050 °C. It should be noted that along with the appearance of α-Al_2_O_3_, FCC reflexes appear, and the contents of both rapidly increase with increasing temperature. At the same time, at 1200 °C, the B2 reflex disappears, pointing to AlNi BCC/B2 decomposition [10,24]. This confirms that the appearance of FCC in the HVOF-deposited coatings is a result of BCC destabilization due to aluminum outflow. Cr_2_O_3_ reflexes appear only when the powder is heated to over 1200 °C. The phases determined at different temperatures are listed in Table 4. 

In [24], when the annealing of gas-atomized AlCoCrFeNi powder was studied, FCC formation was reported as a result of nonequilibrium BCC solid solution decomposition. In the present study, FCC is not an equilibrium phase [25], and its formation is caused by the destabilization of BCC solid solution due to Al outflow to oxide.

### 3.4. Oxidation and Phase Transformations in the Powder during Isothermal Annealing

To observe isothermal oxidation, an annealing temperature of 800 °C was chosen, since at lower temperatures oxidation develops too slowly, and at higher temperatures it develops too fast. Figure 7 presents elemental distribution along the line from the particle surface after powder annealing at 800 °C for 1, 9, and 27 h. The surface has an enlarged concentration of Al and O, but the thin-surface oxide film is not detected by XRD. The upper surface layer with a high Al concentration is followed by an Al-depleted zone. It can be noticed that under prolonged annealing, the particle’s core is also gradually depleted with Al. Concentrations of the other elements close to the surface zone decrease in the order of affinity to oxygen decrease: Cr, Fe, Mo, Co, and Ni. A high concentration of sigma-forming elements, such as Cr, Fe, and Mo, in the subsurface layer leads to the formation of a sigma “shell” around the particle instead of FCC, as was observed in the coatings and continuously heated powder. 

The microhardness of the gas-atomized and annealed-at-800 °C particles is presented in Table 5. The decomposition of the supersaturated BCC solid solution leads to significant strengthening due to hard sigma phase formation. After annealing for 27 h, the microhardness decreases, which can be explained by a significant outflow of Al to the surface oxide and an increased porosity due to the Kirkendall effect. However, even after annealing at 800 °C for 27 h, the particles’ microhardness is superior to that of gas-atomized powder.

Thus, during prolonged isothermal annealing, the decomposition of the supersaturated BCC solid solution with sigma phase formation occurs earlier than oxidation. 

The phase transformation sequence in FeNiCoCrMo_0.5_Al_1.3_ particles during HVOF deposition and annealing in air atmosphere is different. High temperature and high oxygen concentration in spraying lead to oxidation development prior to supersaturated solid solution decomposition. Due to the extremely high rate of heating and cooling during HVOF spraying, the sigma phase has no sufficient time to form. Zones of supersaturated BCC solid solution that are depleted in Al transfer to FCC. In contrast, under annealing, the supersaturated BCC solid solution decomposition occurs earlier than pronounced oxidation, observable with XRD. The decomposition of the BCC supersaturated solid solution with sigma formation occurs in the temperature interval of 600–800 °C, and over 800 °C the pronounced oxidation with alumina formation begins. Alumina growth leads to the appearance of FCC in the zones depleted in Al. During prolonged annealing at 800 °C, the subsurface layer is enriched with Cr, Fe, and Mo, which prevent FCC formation. 

## 4. Conclusions

Oxidation and phase transformations of FeNiCoCrMo_0.5_Al_1.3_ high-entropy alloy powder produced via gas atomization were studied under HVOF deposition, during continuous heating, and prolonged isothermal annealing at 800 °C. The following was found:Under HVOF spraying, the processes of oxidation and phase transformations occur in the following sequence: (i) pronounced Al diffusion from subsurface BCC supersaturated solid solution to the particle surface; (ii) particle’s surface Al_2_O_3_ formation; and (iii) FCC formation in the Al-depleted subsurface layer.During continuous heating in an air atmosphere, processes occur in the following sequence: (i) decomposition of the BCC supersaturated solid solution with sigma formation; (ii) Al diffusion to the surface; and (iii) simultaneous formation of Al_2_O_3_ and FCC due to Al outflow from BCC/B2.During prolonged isothermal annealing at 800 °C in an air atmosphere, processes occur in the following sequence: (i) decomposition of the BCC supersaturated solid solution with sigma formation; (ii) diffusion of Al, Cr, Fe, and Mo to the surface; and (iii) formation of surface Al_2_O_3_ and undersurface sigma due to enrichment of the undersurface zone with Cr, Fe, and Mo.

Thus, FeNiCoCrMo_0.5_Al_1.3_ coatings with high alumina content can be produced via HVOF using a longer distance from the nozzle to the substrate, or a finer powder. In order to prevent FCC formation during deposition, feedstock powder with excessive Al concentration should be used. Preliminary heat treatment of the feedstock powder allows us to obtain powder with a higher hardness and protective alumina shell. Due to a high hardness and high volume of alumina, the FeNiCoCrMo_0.5_Al_1.3_ coatings can be especially useful for tribotechnical application at elevated temperatures.

## Figures and Tables

**Figure 1 materials-17-00531-f001:**
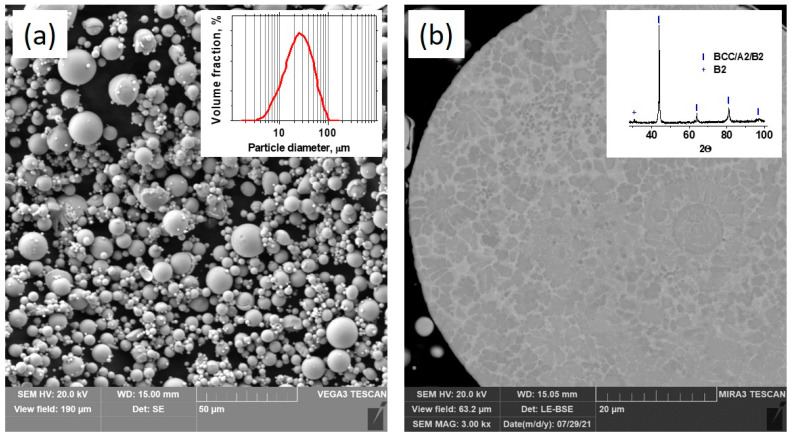
SEM image of gas-atomized FeNiCoCrMo_0.5_Al_1.3_ powder (**a**); cross-sectional SEM image of the particle (**b**); inset of (**a**): particle size distribution; inset of (**b**): XRD pattern of the gas-atomized powder.

**Figure 2 materials-17-00531-f002:**
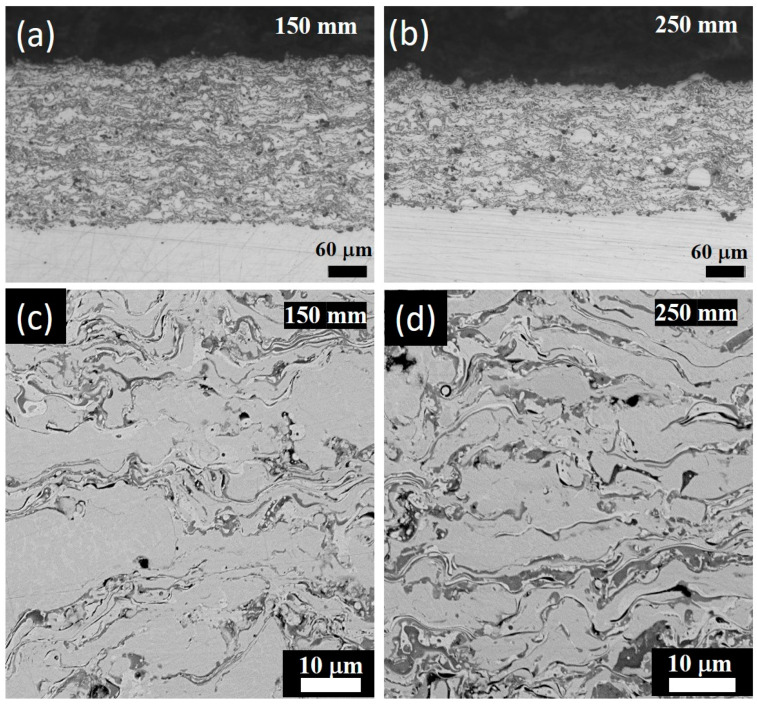
Cross-sectional optical (**a**,**b**) and SEM (**c**,**d**) images of the coatings sprayed using 150 mm (**a**,**c**) and 250 mm (**b**,**d**) distances to the substrate.

**Figure 3 materials-17-00531-f003:**
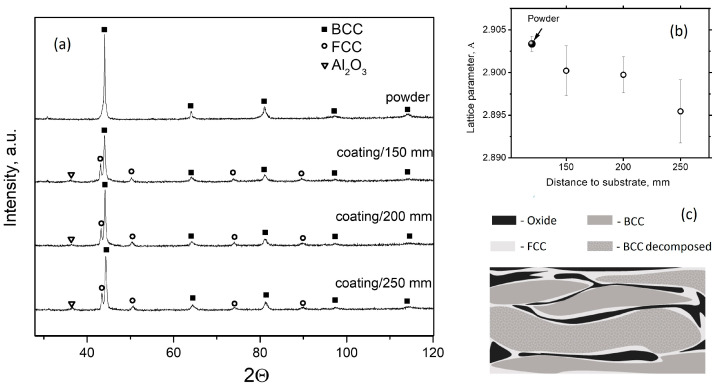
XRD pattern of the gas-atomized powder and the coatings produced using different distances to the substrate (**a**). Lattice parameter of the BCC solid solution based on the distance to the substrate (**b**). Schematic microstructure of the coating (**c**).

**Figure 4 materials-17-00531-f004:**
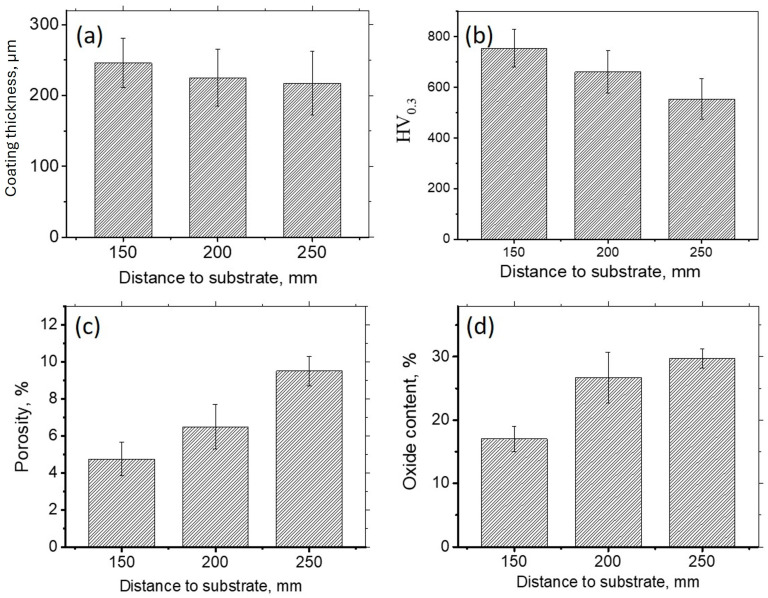
Thickness (**a**), microhardness (**b**), porosity (**c**), and oxide content it is done (**d**) in the coatings made via spraying the substrate from different distances.

**Figure 5 materials-17-00531-f005:**
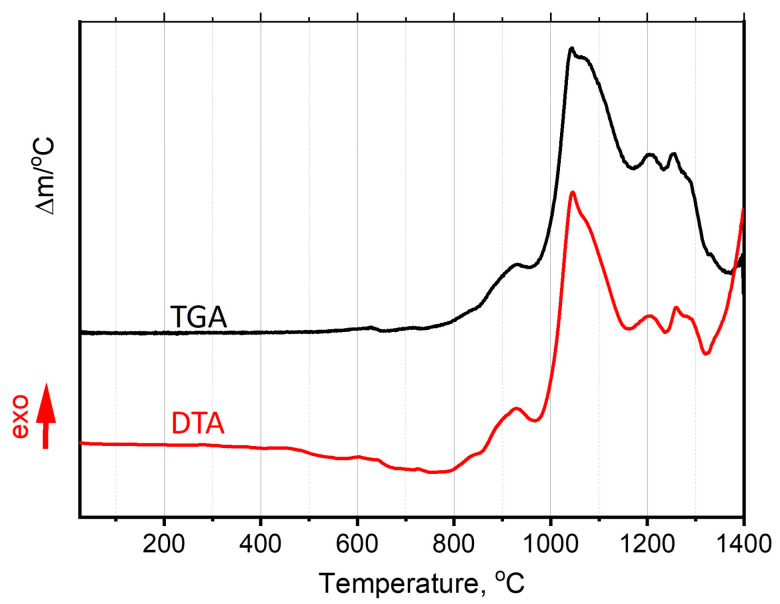
DTA and TGA curves.

**Figure 6 materials-17-00531-f006:**
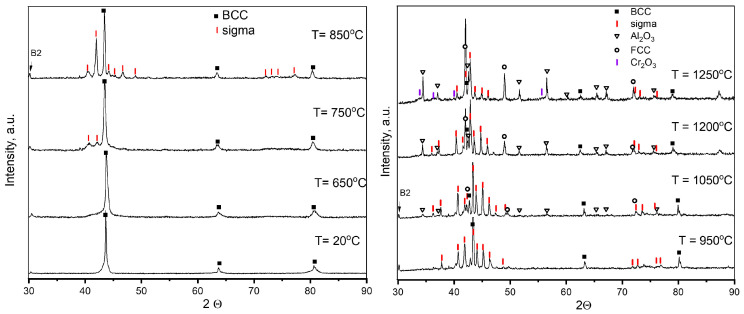
XRD patterns of the powder heated to different temperatures.

**Figure 7 materials-17-00531-f007:**
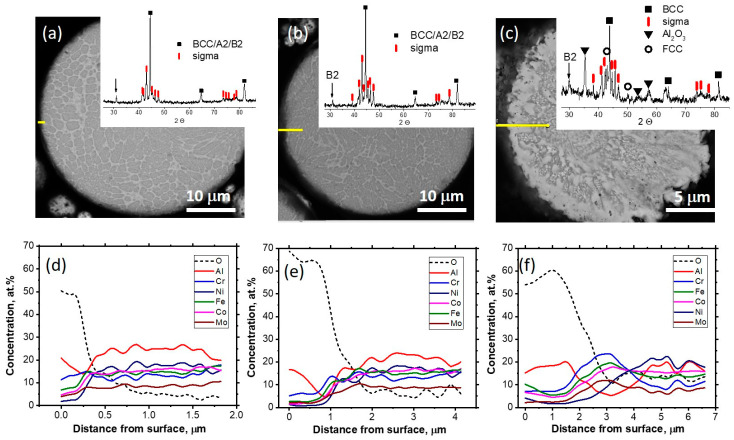
SEM image (**a**–**c**) and EDX line scanning results (**d**–**f**) of the particles annealed at 800 °C for 1 (**a**,**d**), 9 (**b**,**e**), and 27 h (**c**,**f**). EDS scanning line—yellow line in (**a**–**c**). Inset of (**a**–**c**)—XRD patterns.

**Table 1 materials-17-00531-t001:** Alloy atomic concentration, %.

	Atomic Concentration, %					
	Fe	Ni	Co	Cr	Mo	Al
Powder	17.2	17.3	17.2	16.9	9.3	22.0
Coating	17.2	17.8	17.5	16.7	9.0	21.7

**Table 2 materials-17-00531-t002:** HVOF spraying parameters.

Parameter	Value
Oxygen consumption (L/min)	250
Propane consumption (L/h)	30
Nitrogen pressure (kg/cm^2^)	5
Powder feed rate (rpm)	5.2
Spraying distance (mm)	150/200/250
Surface speed (m/min)	5
Layer number	150

**Table 3 materials-17-00531-t003:** Elemental compositions of the phases.

Phase	Atomic Concentration, %					
	Fe	Ni	Co	Cr	Mo	Al
BCC (grey)	17.6	17.5	17.8	17.0	9.2	21.0
FCC (light)	15.6	28.5	26.8	8.9	12.2	8.0
Oxide (dark)	17.2	15.3	16.6	15.6	8.9	26.3

**Table 4 materials-17-00531-t004:** Phase composition of the FeNiCoCrMo_0.5_Al_1.3_ powder heated to different temperatures.

Temperature	Phases
20 °C	BCC/A2; BCC/B2
650 °C	BCC/A2; BCC/B2; sigma (pre-precipitates)
750 °C	BCC/A2; BCC/B2; sigma
950 °C	BCC/A2; BCC/B2; sigma
1050 °C	BCC/A2; BCC/B2; sigma; α-Al_2_O_3_; FCC
1200 °C	BCC/A2; sigma; α-Al_2_O_3_; FCC
1250 °C	BCC/A2; sigma; α-Al_2_O_3_; FCC; Cr_2_O_3_

**Table 5 materials-17-00531-t005:** Microhardness of the FeNiCoCrMo_0.5_Al_1.3_ particles in gas-atomized and annealed conditions.

Condition	HV
gas—atomized	770 ± 50
annealed at 800 °C for 1 h	840 ± 50
annealed at 800 °C for 9 h	910 ± 40
annealed at 800 °C for 27 h	800 ± 50

## Data Availability

Data are contained within the article.

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
