# Peer review of "Oxidation Behavior of FeNiCoCrMo0.5Al1.3 High-Entropy Alloy Powder"

_materials, 2024, doi:10.3390/ma17020531_

Round 1

Reviewer 1 Report

Comments and Suggestions for Authors

In this work, the authors studied the microstructure and oxidation behavior of FeNiCoCrMoAl HEA powder produced with gas atomization. Additionally, the powder has been sprayed with HVOF with the employment of different spray distances. The microstructure of the produced coatings was assessed. This work is of high interest for the field of HEAs and HEA coatings. Please find my comments below:

1) Please expand the introduction explaining the novelty of this work.

2) According to Fig 3, the phase constitution of the rapidly solidified powder (Gas atomized) and the coatings is different. The effect of the different cooling rates on the microstructure of HEAs, both bulk and coatings, is a very important topic. It would be worth expanding the discussion on this discrepancy and explaining the mechanisms. You may use relevant works:

https://doi.org/10.3390/coatings13061004

3) Please expand table 1 and add the chemical composition of the produced coatings. Might be better to move the table to the result section.

4) Why is the microhardness of the powder that was heat treated at 800C at 27h decreasing compared to the powder that was annealed for shorter periods of time? How is this related with the microstructure? Please expand.

5) You need to add conclusions. It would be worth suggesting potential applications for FeNiCoCrMoAl coatings.

Reviewer 2 Report

Comments and Suggestions for Authors

The work is mainly focused on the different transformations occurring in atomized HEA powders when they are sprayed  compared with those occurring during isothermal annealings. The study evidences that selective aluminium oxidation  during spraying changes the nature of the phase transformations occurring in atomized HEA powders. In general, the manuscript is well presented and the data clearly discussed. Nevertheless, there are few unclear points that should be addressed in a new version of the manuscript.

1) There is a significant decrease in hardness with increasing the spraying distance which is mainly associated with an increase of the porosity. In addition, it seems that incorporation of a higher content of oxides has no influence in hardness values. This is strange because the porosity only ranges from 5 to 9 %  as the distance increases but the percentage of oxides changes from 17 to 30 %. This is a significant increase in the volume fraction of hard phases such oxides.  This would suggest an increase in hardness (not observed).  Thus, such behaviour should be associated with the change in the volume fraction of the soft FCC-phase and the Al-depleted BCC-phase. It would be interesting to know the volume fraction of both phases and their corresponding hardness. Probably, the hardness of FCC- and BCC-phases become smaller as the spraying distance is increased.

2) Initially, it is written that the atomized powders consist exclusively of the  disordered BCC-phase, while FCC-phase can appear in the coatings (see Fig. 3). However, the presence of ordered B2-phase is noticed in the coatings from room temperature up to 1050°C. Is the B2 phase present in the coatings? 

3) According to Fig. 6 sigma phase is formed between 650 and 1250°C. Usually, sigma formation is restricted at temperatures below 1000°C. How sigma phase can be stable at temperatures beyond 1000°C?

4) Cr2O3 is detected beyond 1200°C. However, such oxide becomes volatile in oxygen rich atmospheres above 1000-1100°C. Are the authors sure that Cr2O3  is formed during oxidation above 1200°C?

5) In XRD patterns of Fig. 7 no peaks due to Cr2O3  or spinels are present. How can the authors conclude the formation of such oxides during isothermal annealing at 800°C?
